# Ct threshold values, a proxy for viral load in community SARS-CoV-2 cases, demonstrate wide variation across populations and over time

A Sarah Walker[1,2,3,4]*, Emma Pritchard[1,2], Thomas House[5,6], Julie V Robotham[2,7], Paul J Birrell[7,8], Iain Bell[9], John I Bell[10], John N Newton[11], Jeremy Farrar[12], Ian Diamond[9], Ruth Studley[9], Jodie Hay[13,14], Karina-Doris Vihta[1,2], Timothy EA Peto[1,2,3,15], Nicole Stoesser[1,2,3,15†], Philippa C Matthews[1,15†], David W Eyre[1,2,14,16†], Koen B Pouwels[1,2,17], COVID-19 Infection Survey team

[1]Nuffield Department of Medicine, University of Oxford, Oxford, United Kingdom; [2]The National Institute for Health Research Health Protection Research Unit in Healthcare Associated Infections and Antimicrobial Resistance at the University of Oxford, Oxford, United Kingdom; [3]The National Institute for Health Research Oxford Biomedical Research Centre, University of Oxford, Oxford, United Kingdom; [4]MRC Clinical Trials Unit at UCL, UCL, London, United Kingdom; [5]Department of Mathematics, University of Manchester, Manchester, United Kingdom; [6]IBM Research, Hartree Centre, Sci-Tech Daresbury, United Kingdom; [7]National Infection Service, Public Health England, London, United Kingdom; [8]MRC Biostatistics Unit, University of Cambridge, Cambridge Institute of Public Health, Cambridge, United Kingdom; [9]Office for National Statistics, Newport, United Kingdom; [10]Office of the Regius Professor of Medicine, University of Oxford, Oxford, United Kingdom; [11]Health Improvement Directorate, Public Health England, London, United Kingdom; [12]Wellcome Trust, London, United Kingdom; [13]University of Glasgow, Glasgow, United Kingdom; [14]Lighthouse Laboratory in Glasgow, Queen Elizabeth University Hospital, Glasgow, United Kingdom; [15]Department of Infectious Diseases and Microbiology, Oxford University Hospitals NHS Foundation Trust, John Radcliffe Hospital, Oxford, United Kingdom; [16]Big Data Institute, Nuffield Department of Population Health, University of Oxford, Oxford, United Kingdom; [17]Health Economics Research Centre, Nuffield Department of Population Health, University of Oxford, Oxford, United Kingdom

*For correspondence:
sarah.walker@ndm.ox.ac.uk

†These authors contributed equally to this work

## Abstract

**Background:** Information on SARS-CoV-2 in representative community surveillance is limited, particularly cycle threshold (Ct) values (a proxy for viral load).

**Methods:** We included all positive nose and throat swabs 26 April 2020 to 13 March 2021 from the UK's national COVID-19 Infection Survey, tested by RT-PCR for the N, S, and ORF1ab genes. We investigated predictors of median Ct value using quantile regression.

**Results:** Of 3,312,159 nose and throat swabs, 27,902 (0.83%) were RT-PCR-positive, 10,317 (37%), 11,012 (40%), and 6550 (23%) for 3, 2, or 1 of the N, S, and ORF1ab genes, respectively, with median Ct = 29.2 (~215 copies/ml; IQR Ct = 21.9–32.8, 14–56,400 copies/ml). Independent predictors of lower Cts (i.e. higher viral load) included self-reported symptoms and more genes detected, with at most small effects of sex, ethnicity, and age. Single-gene positives almost

invariably had Ct > 30, but Cts varied widely in triple-gene positives, including without symptoms. Population-level Cts changed over time, with declining Ct preceding increasing SARS-CoV-2 positivity. Of 6189 participants with IgG S-antibody tests post-first RT-PCR-positive, 4808 (78%) were ever antibody-positive; Cts were significantly higher in those remaining antibody negative.
**Conclusions:** Marked variation in community SARS-CoV-2 Ct values suggests that they could be a useful epidemiological early-warning indicator.

**Funding:** Department of Health and Social Care, National Institutes of Health Research, Huo Family Foundation, Medical Research Council UK; Wellcome Trust.

## Introduction

After initial reductions in SARS-CoV-2 cases in mid-2020, following release of large-scale lockdowns (*Flaxman et al., 2020*), infection rates have undergone waves of resurgence and suppression in many countries worldwide. Proposed control strategies include new local or national lockdowns of varying intensity and mass testing, but these have major economic and practical limitations. In particular, mass testing of large numbers without symptoms (*Yokota et al., 2020*), and hence low pre-test probability of positivity, can mean most positives are false-positives depending on test specificity. For example, with 0.1% true prevalence, testing 100,000 individuals with a 99.9% specific test with perfect sensitivity gives 100 true-positives, but also 100 false-positives (positive predictive value [PPV] 50%), whereas specificity of 99.5% increases false-positives to 500 (PPV = 17%), and of 99.0% to 999 (PPV = 9%), with even lower PPV with imperfect sensitivity (*Adams et al., 2020*).

Mathematical models are powerful tools for evaluating the potential effectiveness of different control strategies, but rely on population-level estimates of infectivity and other parameters. However, there are few unbiased community-based surveillance studies, including individuals both with and without symptoms. Estimates of asymptomatic infection rates vary, being 17–41% overall in recent reviews (*Buitrago-Garcia et al., 2020*; *Byambasuren et al., 2020*), but these included many studies of contacts of confirmed cases. Higher prevalence of asymptomatic infection has been reported in screening of defined populations (30% [*Buitrago-Garcia et al., 2020*]) and community surveillance (e.g. 42% *Lavezzo et al., 2020*, 72% *Riley and Ainslie, 2020a*). Studies have generally indicated lower rates of transmission from asymptomatic infection (*Buitrago-Garcia et al., 2020*; *Byambasuren et al., 2020*), this may be a proxy for SARS-CoV-2 viral load as a key determinant of transmission. Finally, most studies rely on 'average' estimates of the asymptomatic infection percentage, independent of characteristics and viral load, and have not quantified temporal variation in these key parameters for mathematical models across the community.

Here we therefore characterise variation in SARS-CoV-2-positive tests in the first 11 months of the UK's national COVID-19 Infection Survey. In brief (details in Materials and methods), the survey randomly selects private households to provide a representative UK sample, recruiting all consenting individuals aged 2 years or older currently resident in each household to provide information on demographics, symptoms, contacts and relevant behaviours and self-taken nose and throat swabs for RT-PCR testing (*Pouwels et al., 2021*). A randomly selected subset is approached for additional consent to provide blood samples for IgG S-antibody testing if aged 16 years or older. At the first visit, participants can provide additional consent for longitudinal follow-up (visits every week for the next month, then monthly for 12 months from enrolment). We estimate predictors of RT-PCR cycle threshold (Ct) values (as a proxy for viral load), propose a classification for the strength of evidence supporting positive RT-PCR test results in the community, and demonstrate how this has changed over time. We also provide a preliminary assessment of seroconversion rates for community cases.

## Materials and methods

This study included all positive SARS-CoV-2 RT-PCR results between 26 April 2020 and 13 March 2021 from nose and throat swabs taken from participants in the Office for National Statistics (ONS) CIS (ISRCTN21086382). The survey randomly selects private households on a continuous basis from address lists and from previous surveys to provide a representative UK sample (*Supplementary file 1*). If anyone aged 2 years or older currently resident in an invited household agreed verbally to participate, a study worker visited the household to take written informed consent, which was obtained

from parents/carers for those 2–15 years; those aged 10–15 years provided written assent. The study protocol is available at https://www.ndm.ox.ac.uk/covid-19/covid-19-infection-survey/protocol-and-information-sheets. Recruitment started 26 April 2020 in England, 29 June 2020 in Wales, 29 July 2020 in Northern Ireland, and 21 September 2020 in Scotland.

Individuals were asked about demographics, symptoms, contacts, and relevant behaviours (https://www.ndm.ox.ac.uk/covid-19/covid-19-infection-survey/case-record-forms). To reduce transmission risks, self-taken nose and throat swabs were obtained following study worker instructions. Parents/carers took swabs from children under 12 years. At the first visit, participants were asked for (optional) consent for follow-up visits every week for the next month, then monthly for 12 months from enrolment. In a random 10–20% households, those 16 years or older were invited to provide venous blood monthly for assays of anti-trimeric spike protein IgG using an immunoassay developed by the University of Oxford (*National SARS-CoV-2 Serology Assay Evaluation Group, 2020*). All participants in households where anyone tested positive on a swab were also invited to provide blood monthly. Venous blood was not taken at any visit where any person in the household had classic COVID-19 symptoms (fever, cough, or anosmia/ageusia). The study received ethical approval from the South Central Berkshire B Research Ethics Committee (20/SC/0195).

Swabs and blood samples were collected by study workers at household visits and couriered overnight to testing laboratories at ambient temperatures. They were analysed at the UK's national Lighthouse Laboratories at Milton Keynes (National Biocentre) (from 26 April 2020 to 11 February 2021) and Glasgow (from 16 August 2020) using identical methodology, with swabs from specific regions sent consistently to one laboratory. RT-PCR for three SARS-CoV-2 genes (N protein, S protein, and ORF1ab) used the Thermo Fisher TaqPath RT-PCR COVID-19 Kit, analysed using UgenTec Fast Finder 3.300.5 (TaqMan 2019-nCoV Assay Kit V2 UK NHS ABI 7500 v2.1). The Assay Plugin contains an Assay-specific algorithm and decision mechanism that allows conversion of the qualitative amplification Assay PCR raw data from the ABI 7500 Fast into test results with minimal manual intervention. Samples are called positive in the presence of at least single N gene and/or ORF1ab but may be accompanied with S gene (one, two, or three gene positives). There is no specific Ct threshold for determining positivity. S gene is not considered a reliable single-gene positive (as of mid-May 2020). Blood was analysed at the University of Oxford. Antibody titres were considered positive above 8 million units (*National SARS-CoV-2 Serology Assay Evaluation Group, 2020*) on the original fluorometric version of the assay and 42 units on the colorimetric version of the assay (used from 1 March 2021).

Twelve specific symptoms were elicited at each visit (cough, fever, myalgia, fatigue, sore throat, shortness of breath, headache, nausea, abdominal pain, diarrhoea, loss of taste, loss of smell), as was whether participants thought they had (unspecified) symptoms compatible with COVID-19. From 26 April through 22 July 2020, questions referred to current symptoms, and from 23 July 2020 to the preceding 7 days. Any positive response to any symptom question at the swab-positive visit defined the case as symptomatic 'at' the test; we also separately defined any positive response at the swab-positive visit or visits either side (regardless of time between visits) as symptomatic 'around' the test.

To investigate the potential increasing contribution of false-positives as population prevalence declines, from 2 August 2020 we arbitrarily classified in real-time each positive as:

- 'Higher' evidence: two or three genes detected (irrespective of Ct).
- 'Moderate' evidence: single-gene detected and (1) Ct below the 97.5th percentile of 'higher' evidence positives (<34; supporting this threshold, whole genome sequences had been obtained from three single-gene positives with Ct 30.8–33.1 by 2 August) or (2) higher pre-test probability of infection, defined as any symptoms at/around the test or reporting working in a patient-facing healthcare or care/residential home.
- 'Lower' evidence: all other positives; by definition single-gene detected at $Ct \geq 34$ in individuals not reporting symptoms/working in relevant roles.

As the Ct distribution was skewed to the left, we assessed independent predictors using median (quantile) regression. Results were broadly similar using random effects model for mean Ct with a random effect per household. We used five knot natural cubic splines (knots at the 10th/25th/50th/75th/90th percentiles of observed unique values) to assess non-linearity in the effect of calendar time, age, and deprivation (index of multiple deprivation rank). Multivariable models for Ct values

were constructed by first choosing the more strongly univariably predictive factor from the collinear variables (symptoms at/around the test, number of genes detected/supporting evidence for each positive) and then using backwards elimination on the remaining variables. Deprivation was assessed using the index of multiple deprivation (IMD) in England, a score based on lower layer super output areas with average population of 1500 people and incorporating seven domains to produce an overall relative measure of deprivation (income, employment, education, skills and training, health and disability, crime, barriers to housing services and living environment) (https://www.gov.uk/government/statistics/english-indices-of-deprivation-2019) and equivalent scores in the other three countries comprising the UK. Each country's scores were converted to a within-country percentile. All analyses were conducted in Stata 16.1.

## Results

### Number and percentage of positive swabs

From 26 April 2020 to 13 March 2021, 440,479 participants from 217,887 households in the COVID-19 Infection Survey had one or more RT-PCR results from nose and throat swabs (median eight results per participant [IQR 6–9, range 1–19]). Participants were recruited between April 2020 and March 2021 (*Supplementary file 1*). Of 3,312,159 RT-PCR test results, 27,902 (0.84%, 95% CI 0.83–0.85%) were positive, in 21,831 individuals from 16,214 households. Two thousand nine hundred and sixty-six (14%) of these individuals were positive at their first test in the study and 18,865 (86%) subsequently, after median five negative tests (IQR 3–6, range 1–14).

### Viral characteristics

Overall, 10,317 (37%), 11,012 (40%), and 6550 (23%) swabs were positive for three, two, or one of the three SARS-CoV-2 genes (N protein, S protein, and ORF1ab), respectively (*Table 1*; 23 positives with missing Ct and gene detection excluded from this and all subsequent analysis; samples with only the S-gene detected generally not called positive, see Materials and methods). The majority of two-gene positives (9513 [86%]) were ORF1ab+N positive from 16 November 2020 onwards, reflecting the emergence and expansion of B.1.1.7 (WHO Alpha) in the UK (*Walker et al., 2021*). B.1.1.7

**Table 1.** Genes detected in positive swabs.

| Number of genes detected | All positives (N = 27,879) | | First positive per participant (N = 21,811) | |
|---|---|---|---|---|
| | N (%) | Median CT* (IQR) [range] | N (%) | Median CT* (IQR) [range] |
| 1 | 6550 (23%) | 33.8 (32.9–34.7) [12.7–38.7] | 5102 (23%) | 33.9 (32.9–34.7) [12.7–38.7] |
| 2 | 1145 (4%) | 32.3 (30.9–33.4) [10.3–37.2] | 773 (4%) | 32.3 (30.7–33.4) [10.3–37.2] |
| 2: ORF1ab+N 16 Nov 2020 onwards | 9867 (35%) | 26.4 (19.4–31.1) [9.2–37.8] | 8184 (38%) | 25.3 (18.6–30.7) [9.2–37.8] |
| 3 | 10,317 (37%) | 25.3 (19.8–29.5) [9.3–36.8] | 7752 (36%) | 23.9 (18.8–28.8) [9.3–36.8] |
| **Genes detected** | | | | |
| N only | 4479 (13%) | 33.9 (33.0–34.8) [26.1–38.7] | 3419 (16%) | 34.0 (33.1–34.8) [28.2–38.7] |
| ORF1ab only | 2044 (7%) | 33.6 (32.6–34.5) [16.8–38.3] | 1656 (8%) | 33.7 (32.7–34.6) [16.8–38.3] |
| S only† | 27 (0.1%) | 34.9 (33.5–36.1) [12.7–37.3] | 27 (0.1%) | 34.9 (33.5–36.1) [12.7–37.3] |
| N+ORF1ab: before 16 Nov 2020 | 731 (3%) | 31.9 (30.3–32.9) [10.3–37.2] | 497 (2%) | 31.8 (29.7–33.0) [10.3–38.2] |
| N+ORF1ab: 16 Nov 2020 onwards | 9867 (35%) | 26.4 (19.4–31.1) [9.2–37.8] | 8184 (38%) | 23.9 (18.8–28.8) [9.3–36.8] |
| S+ORF1ab | 190 (0.7%) | 32.5 (31.2–33.5) [15.1–36.6] | 138 (0.6%) | 32.4 (31.0–33.6) [15.1–36.6] |
| N+S | 224 (0.8%) | 33.4 (32.5–34.2) [25.0–36.8] | 138 (0.6%) | 33.3 (32.4–34.3) [27.3–36.8] |
| N+S+ORF1ab | 10,317 (37%) | 25.3 (19.8–29.5) [9.3–36.8] | 7752 (36%) | 25.3 (18.6–30.7) [9.2–37.8] |

*Taking the mean Ct per positive swab across positive gene targets (Spearman rho = 0.98 for each pair of genes where both positive, p<0.0001).

†17/27 before mid-May only: after this samples positive for the S gene only were not called positive overall by the algorithm and therefore reflect likely recording errors.

Note: excluding 23 positive results without Ct values or genes detected available. Comparing first vs subsequent positives per participant, exact p<0.0001 for both number of genes detected and specific genes detected.

leads to S-gene target failure (SGTF) and was estimated to account for 88% of SGTF from this time (***Public Health England, 2020***). Where multiple genes were detected, the Cts were highly correlated (Spearman rho = 0.98, p<0.0001). Taking the per-swab mean Ct across positive genes, the overall median Ct was 29.2 (IQR 21.9–32.8; range 9.2–38.7), reflecting the study's surveillance design testing individuals in the community at fixed timepoints regardless of symptoms. Based on calibration data (***Appendix 1—figure 1***), this corresponds to a median viral load of ~215 copies/ml (IQR 14–56,400). Ct varied strongly by number of genes detected (Kruskal–Wallis p=0.0001), but not by their specific pattern after adjusting for number (p=0.08). There is no fixed Ct threshold for determining positivity (see Materials and methods); however, only 38 (0.1%) Ct values > 37 were recorded (five positive on ORF1ab+N).

Of note, whilst single-gene positives almost invariably had Ct>30, with or without reported symptoms, triple-gene positives without reported symptoms had widely varying Ct, as did ORF1ab+N positives after 16 November 2020 (SGTF, compatible with B.1.1.7) (***Figure 1***). Ct values were slightly but significantly lower in other double-gene positives vs single-gene positives, with a small number of low Ct values in ORF1ab+N positives before 16 November 2020 likely reflecting early B.1.1.7 cases. Furthermore, whilst the percentage reporting symptoms increased linearly as Ct values dropped from 35 (~30% reporting symptoms around the positive test) to 28 (~60% reporting symptoms), below 28 the percentages reporting symptoms increased only slightly (to ~70% at Ct=10) (***Figure 2***).

## Evidence supporting positive results

Combining information on Ct values, symptoms and pre-test probability of being positive, 21,329 (77%), 4741 (17%), and 1809 (6%) positive tests had 'higher', 'moderate', or 'lower' evidence supporting genuine presence of viral RNA (***Table 2***; definitions in Materials and methods). Even though 'higher' evidence was based only on number of genes detected (two or three), 'higher' evidence positives were more likely to be symptomatic than 'moderate' evidence positives (p<0.0001), but

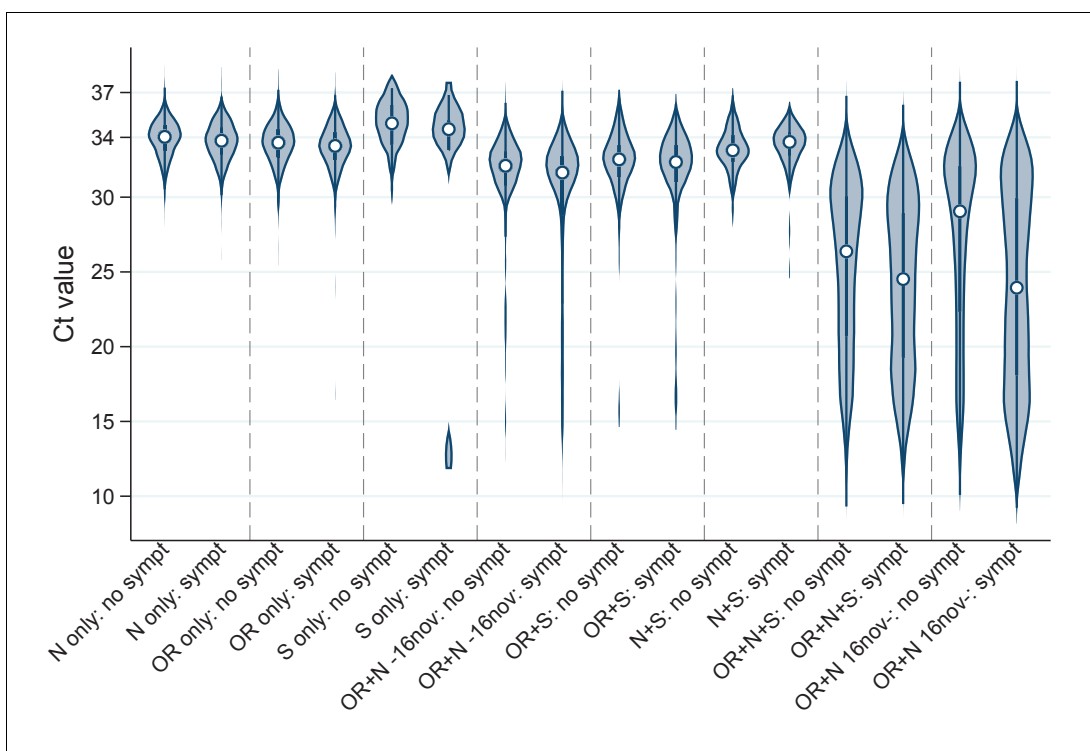

**Figure 1.** Distribution of Ct values at each positive test by genes detected and self-reported symptoms. Note: Points show the median and boxes the interquartile range. OR=ORF1ab. Positives where only the ORF1ab+N genes were detected are split by whether the swab was taken before or after 16 November 2020, reflecting the expansion of B.1.1.7 (which has S-gene target failure on the assay used in the survey).

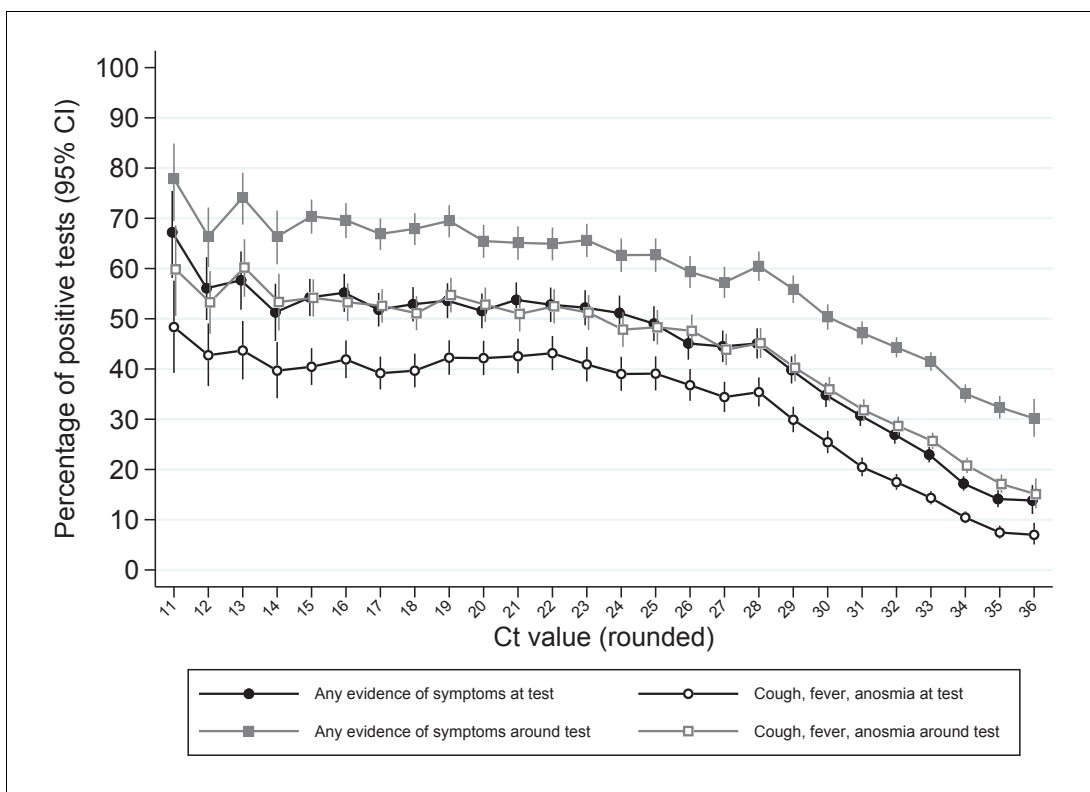

**Figure 2.** Percentage reporting symptoms by Ct value. Note: Points show the percentage of positive tests with each rounded Ct value reporting any symptoms or cough, fever, anosmia/ageusia at each test or around each test (see Materials and methods for symptoms collection and definitions). Ct values under 11 and over 36 grouped with 11 and 36, respectively.

were similarly likely to have occupational risk factors (p=0.48). 'Higher' evidence positives were more likely to occur in households with other positives (p<0.0001).

## Predictors of Ct values

In multivariable regression models, Ct values were independently lower (i.e. viral loads higher) with more genes detected (8.2 lower in triple-gene vs single-gene positives [95% CI 7.9–8.5]), if symptoms were reported around the test (2.0 lower [1.8–2.2]), at the first positive identified per participant (2.2 lower than subsequent positives [2.2–2.5]), and if the positive was not the participant's first test in the study (0.6 lower [0.2–0.9]) (all p<0.0001; *Supplementary file 2A*; see Materials and methods for details of collection of symptoms). By far the strongest effect was associated with triple-gene positives. Men had slightly lower Ct values than women (0.3 lower [0.1–0.5] p=0.001), and there was marginal evidence of lower values in those reporting non-white ethnicity (0.3 lower [0–0.6] p=0.08). Compared with those not reporting symptoms, Ct values were lower in those reporting cough/fever/anosmia/ageusia (2.5 lower [2.3–2.8]) than other symptoms only (0.9 lower [0.7–1.2]; heterogeneity p<0.0001). Associations were similar for symptoms at the positive test. After adjusting for these factors, there was no evidence of independent effects of age (p=0.33) or deprivation (p=0.67, *Supplementary file 2A*). Even after adjusting for these factors, Ct values were 1.4 (1.2–1.6) lower in individuals where another household member was positive at any point in the study (p<0.0001; other effects similar).

However, number of genes detected and symptoms are both potential mediators of effects of demographic factors (*Appendix 1—figure 2*). Excluding these potential mediators (number of genes detected, symptoms), Ct values remained independently lower (i.e. viral loads higher) at the first positive identified per participant, where the positive was not the participant's first test in the study, and in men, but were also slightly lower with increasing deprivation (p=0.0005; Ct 1.0 lower in the most vs least deprived [95% CI 0.6–1.5]) and in younger adults (p=0.0001; those aged 17–24 1.0

**Table 2.** Evidence supporting positive test results indicating presence of virus and impact on other factors.

| | Strength of evidence for true infection | | | |
| --- | --- | --- | --- | --- |
| | Higher | Moderate | Lower | p (exact) |
| Number (col %) (N = 27,879) | 21,329 (77%) | 4741 (17%) | 1809 (6%) | |
| Factors determining classification | | | | |
| Number of genes detected (row %) | 3: 10,317 (48%) 2: 11,012 (52%) | 1: 4741 (100%) | 1: 1809 (100%) | |
| CT, median | 26.2 | 33.4 | 34.8 | |
| CT, n (row %) <34* | 21,070 (98.8%) | 3613 (76%) | 0 (0%) | |
| Symptoms around test, n (row %) | 12,466 (58%) | 2243 (47%) | 0 (0%) | <0.0001 (exc lower) |
| Occupational risk[†], n (row %) | 1322 (6%) | 307 (6%) | 0 (0%) | 0.48 (exc lower) |
| Other factors | | | | |
| Cough, fever, anosmia, ageusia around test, n (row %) | 9345 (44%) | 1241 (26%) | 0 (0%) | <0.0001 (exc lower) |
| First positive test n (row %) (vs subsequent positive test) | 16,709 (78%) | 3508 (74%) | 1594 (88%) | <0.0001 |
| First test in study, n (row %) (vs follow-up i.e. prior negative in study) | 2281 (11%) | 482 (10%) | 199 (11%) | 0.49 |
| Any genome sequence obtained, confirming presence of virus‡ | 6,621/9,022 (73%) | 544/2,315 (24%) | 0/836 (0%) | <0.0001 |
| Any other household member ever positive[$] | 11,493/18,494 (62%) | 1,513/4,004 (38%) | 318/1,525 (21%) | <0.0001 |

*Approximate 97.5th percentile of CT in higher evidence positives through 2 August 2020 when classification first applied.

†Reported working in a patient-facing healthcare role/care/residential home.

‡Any genome sequence obtained out of attempted (other positives not found or not yet attempted).

$Denominator households with two or more study participants.

Note: Classification arbitrarily determined on 2 August 2020 based on the number of genes detected, Ct values and pre-test probability (see Materials and methods).

lower [0.3–1.7] than those under 12, and 1.4 lower [0.8–2.0] than those aged 70+) (*Supplementary file 2B*). Results were similar adjusting for date of the positive test.

## Temporal changes in Ct values, evidence, and symptomatic percentages

There were strong effects of calendar time on the distribution of Ct values (*Figure 3A,B*), the percentages self-reporting symptoms, or cough/fever/anosmia/ageusia (*Figure 3C*), and strength of evidence supporting each positive result (*Figure 3D*; all p<0.0001). In particular, Ct values were markedly higher in July–August 2020 when population positivity rates were low, with correspondingly very low percentages with symptoms at/around positive tests, and more 'lower' evidence positives. Decreases in Ct values in late August/early September and December 2020 coincided with increases in percentages reporting symptoms and of 'higher' evidence positives, and, in England (*Figure 3B*), with initial rises in official estimates of positivity rates (*Office for National Statistics, 2021*) after very low rates in July/early August 2020, and with much stronger rises in December 2020 (expansion of B.1.1.7). Ct levels rose, and correspondingly percentages reporting symptoms and of 'higher' evidence positives declined, as positivity peaked during November 2020 and January 2021 lockdowns.

However, even within 'higher' evidence positives, median Ct varied strongly over time being higher in July/early August 2020 and after November 2020 and January 2021 lockdowns (*Figure 4A*). 'Lower' evidence positives also formed a larger percentage of all tests during July/early August 2020, despite overall positivity rates being very low (e.g. 0.022% in the 3 weeks starting 20 July 2020; *Figure 4B*). However, interestingly, from September 2020, the percentage of 'lower' evidence positives increased proportionately with 'moderate' and 'higher' evidence positives (*Figure 4B*). The lowest non-zero observed rate of 'low evidence' positives was 0.005% (both in early June and late August), providing an upper bound on the rate of false-positives as defined by identifying virus when none present.

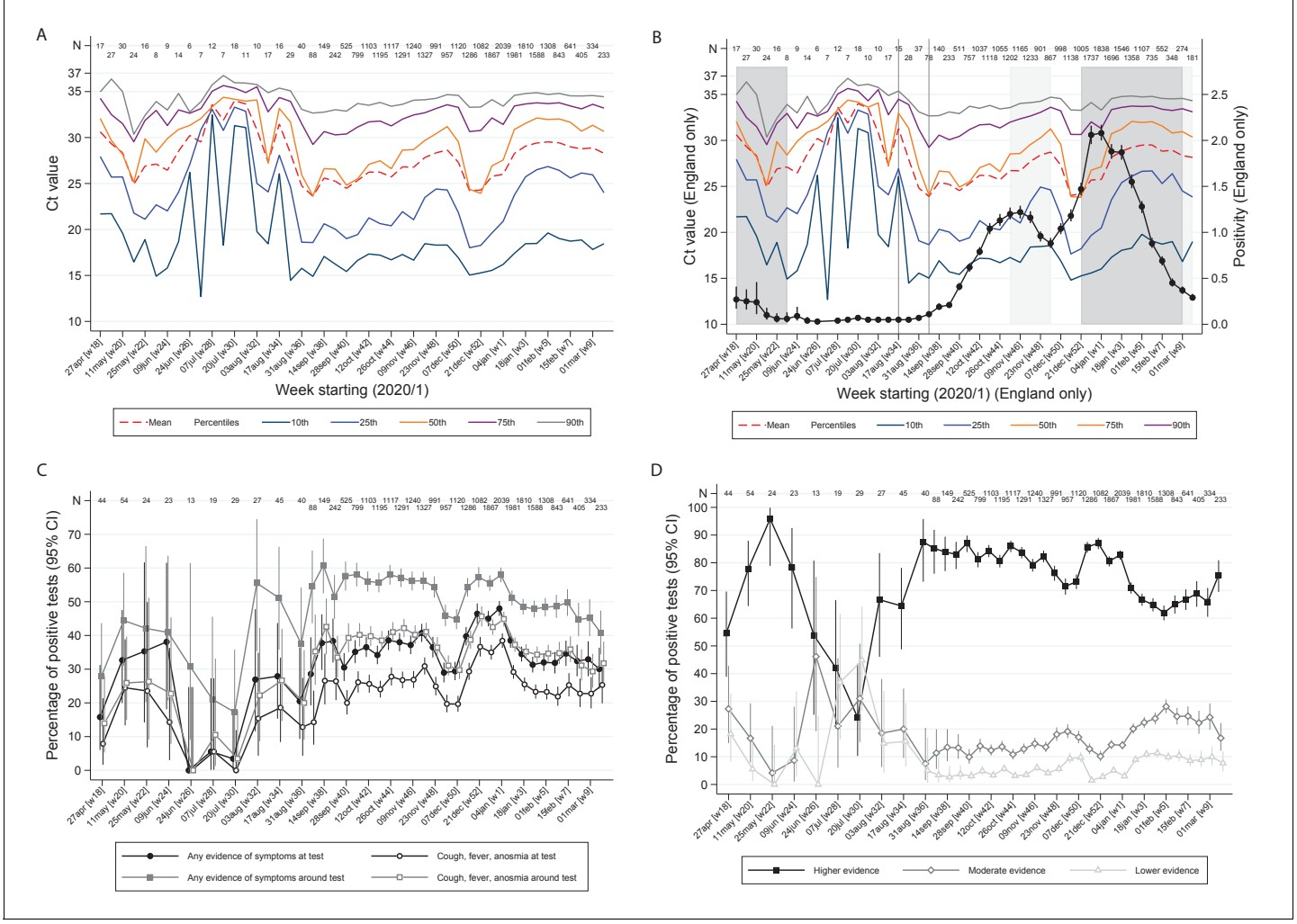

**Figure 3.** Variation over calendar time in the distribution of Ct values in the UK (**A**) and England (**B**) together with percentage positivity in England (**B**), and in self-reported symptoms (**C**) and evidence supporting positives (**D**). Note: Panel (**A**) shows the distribution of Ct values each week including all positives across the UK. Panel (**B**) is restricted to England shown together with the official estimates of positivity as reported by the Office for National Statistics (black line) and periods of national 'stay-at-home' restrictions (schools shut in dark grey, schools open in light gray). Panels (**C**) and (**D**) show the proportions reporting symptoms and with different levels of evidence supporting the positive test, respectively. Variation in the width of 95% CI reflects the increase in size of the survey from mid August (**Supplementary file 1**).

## Relationship with serostatus

One or more IgG S-antibody results were available for 6540 (30%) participants with positive swabs. Less than 5% of antibody tests taken >30 days before the first positive swab (not necessarily the onset of infection) were positive (**Figure 5**), rising to 12% in the 30 days before the first swab positive (likely reflecting late detection of infection), 47% in the following 14 days, and then 72–81% thereafter. Overall, of 6189 participants with one or more antibody tests after their first positive swab, 4808 (78%) were ever antibody-positive; with higher rates in those reporting symptoms around their first positive swab (2945/3315 [89%] vs 1863/2874 [65%] of those not reporting symptoms, p<0.0001). Median (IQR) Ct values were also significantly lower in those ever antibody-positive to date (24.9 [18.5–31.0] vs 33.0 [29.9–34.3] in those not antibody-positive, p<0.0001). Results were similar restricting to 1477 (24%) with a negative antibody result within [−120, +21] days of their first positive swab. A small number of participants appeared to have become infected despite antecedent high anti-spike antibody titres, one case in particular which had 'higher evidence' positive swab tests separated by four consecutive negative swabs with 65 days between positive swabs.

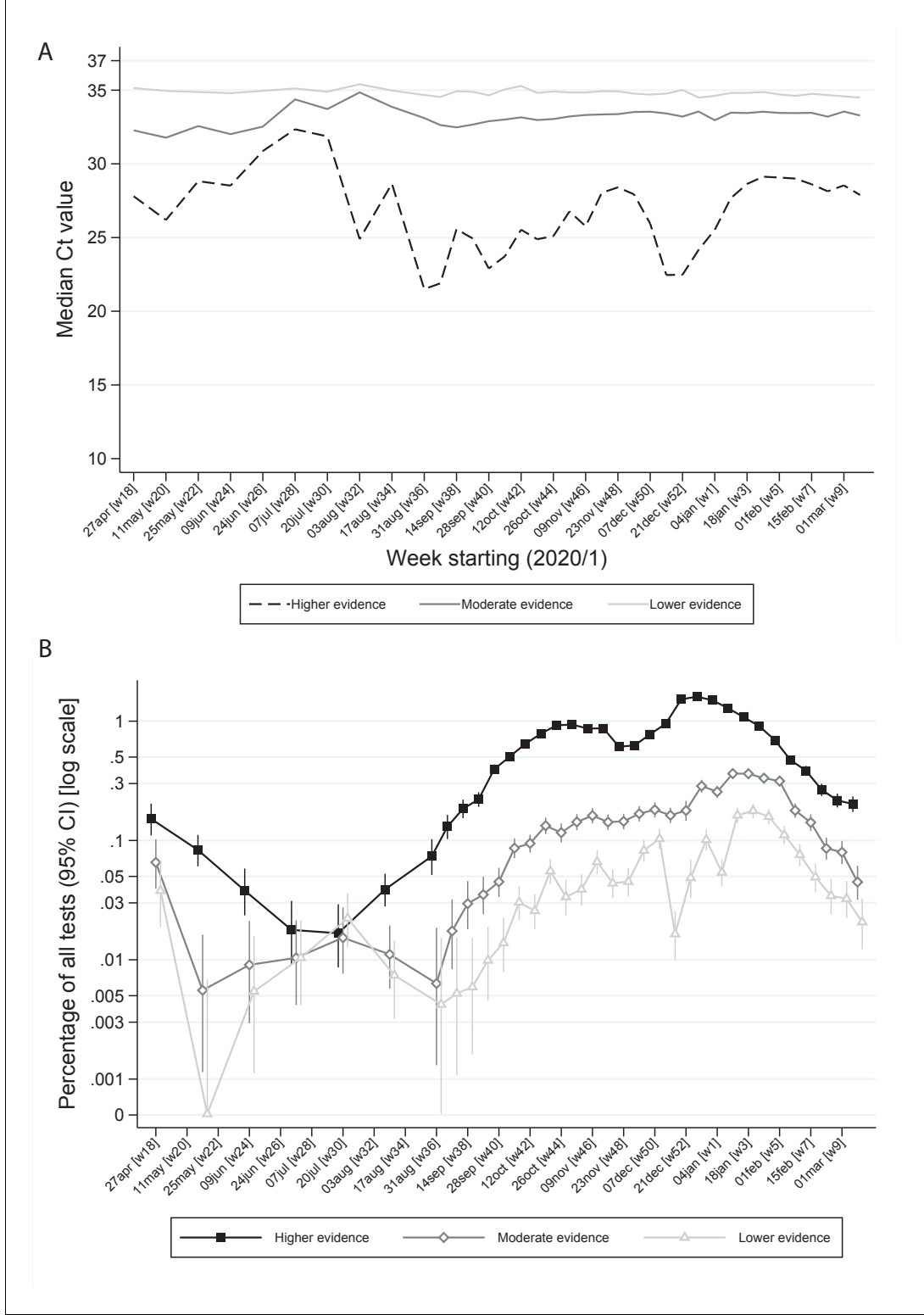

**Figure 4.** Ct values (**A**) and percentage positive of all tests (**B**) by level of evidence and time. Note: Panel (**A**) shows median Ct values according to level of evidence and panel (**B**) percentage of all swab tests positive according to level of evidence over calendar time. The early part of the study is grouped into 3 week periods due to lower numbers of positives.

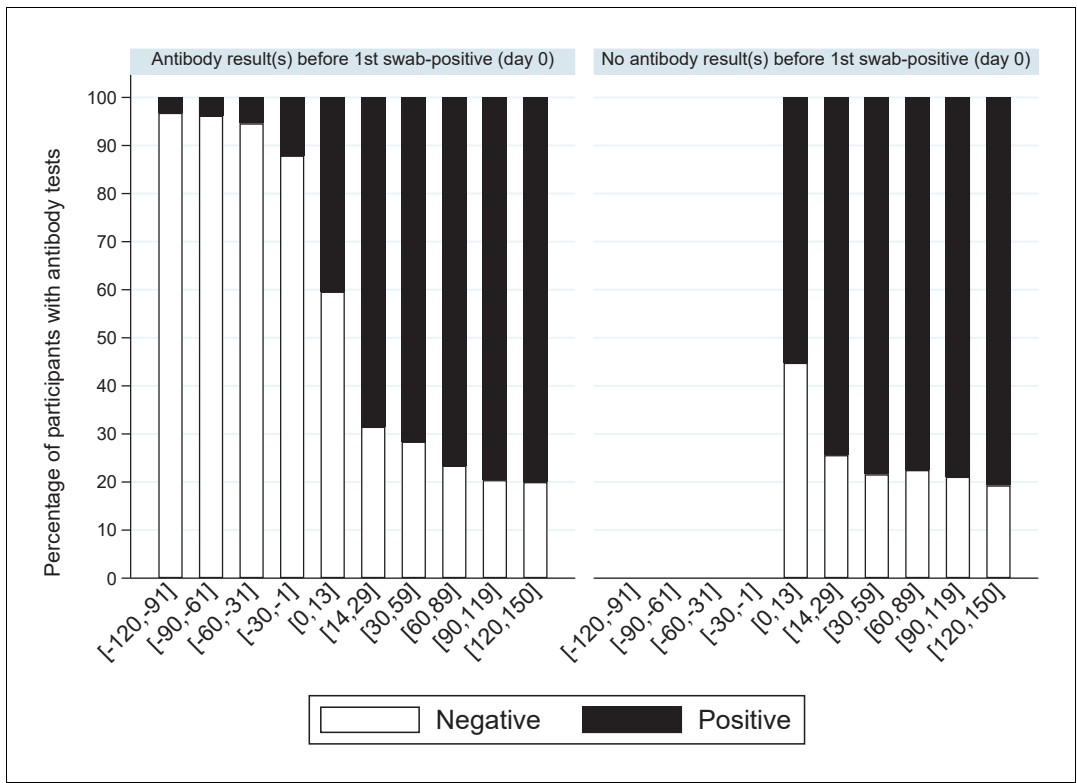

**Figure 5.** Percentage of positive antibody tests over time from first positive swab. Note: showing the percentage of participants with S-antibody positive or negative tests according to days from their first positive swab, separately for those with and without any antibody results prior to their first positive swab.

## Discussion

In this large community surveillance study, we found wide variation in Ct values (a proxy for viral load). Whilst Ct values were independently associated with several factors, including symptoms at/around the test as previously reported (*Edwards et al., 2020*; *Lee et al., 2020*), their effects were small compared with population-level variability. Notably both triple-gene positives and S-gene target failures compatible with the Alpha/B.1.1.7 variant without reported symptoms had widely varying Ct, including many with low values (*Figure 3A*), potentially explaining variation in dispersion ('k') and super-spreading events, particularly from those without symptoms but with low Ct/high viral loads (*Endo et al., 2020*; *Rasmussen and Popescu, 2021*).

Compared with other single/double positives, Ct values were significantly lower in triple-gene positives and S-gene target failures compatible with the B.1.1.7 variant (after mid-November 2020). However, direct comparisons of viral load with B.1.1.7 vs other variants are not possible within this analysis, given lack of knowledge as to the true underlying variant over the included time period. We found lower Ct in those reporting cough/fever/anosmia/ageusia than other symptoms, and other symptoms vs no symptoms, supporting the importance of the 'classic' symptoms for identifying the most infectious cases. Lower Ct values in the first positive per participant likely reflects the natural history of viral load post-infection, and higher Ct values in those positive at their first test in the study over-representation of long-term shedders in this group. Lower Ct values in men and those reporting non-white ethnicity, although small, are consistent with poorer outcomes in these groups. Interestingly, small effects of age and deprivation were mediated by self-reported symptoms and number of positive genes. That is, when adjusting for these latter factors no association was observed with Ct, but without adjustment younger individuals (as shown in *Jones et al., 2020* but not *Jacot et al., 2020*) and those from more deprived areas had slightly lower Ct values, suggesting that these factors may affect the intrinsic level of virus present (*Appendix 1—figure 2*). The small size of the effects mean they may variably be detected depending on study size and power.

Ct values varied strongly over time, as did symptoms and evidence supporting positives, suggesting changing viral burden in infection cases, with less severe infections during July/early August 2020. This strongly refutes hypotheses that declines in positivity during this period were due to declines in viral fitness. During this time, higher Ct values were also noted in the English point-prevalence surveillance study, REACT (*Riley and Ainslie, 2020b*), and lower virus levels in Lausanne, Switzerland (*Jacot et al., 2020*). However, Ct values were higher even in 'higher' evidence positives during this period, consistent with shifting viral burden (*Figure 4A*). Such a shift may also explain the preceding shift towards 'moderate' evidence positives and the concurrent higher percentage of 'lower' evidence positives, since the less virus present, the less likely it is to be detected on multiple genes. Whilst these findings are consistent with lower viral inoculum during this period (*Gandhi et al., 2020*), we cannot assess whether this is predominantly due to behaviour (e.g. increased time outdoors, face mask use *Gandhi and Rutherford, 2020*) or other reasons (e.g. environmental/climatic factors, including relating to transport of swabs for testing). Whilst decreases in Ct values in July/early August 2020 preceded increases in positivity rates in England, later declines in Ct in early December coincided with, rather than preceded, increases in positivity due to B.1.1.7 expansion. This may potentially reflect faster transmission of B.1.1.7 but may also reflect greater sensitivity to changes in Ct distribution when case numbers are small. Subsequent increases in Ct reflected stabilising and then declining positivity in both periods.

We used laboratory, clinical, and demographic evidence to classify our confidence in positive results. Around 70% had two or three genes detected ('higher' evidence), providing assurance in overall results, with only 0.1% of Ct values over 37. Whilst Ct values are not directly comparable between studies, REACT has also validated a Ct threshold of 37 for single-gene positives for their test performed in Germany (*Riley and Ainslie, 2020b*), and in the Public Health England (PHE) Schools study, only samples with Ct<37 were positive on repeat testing of the same swab at PHE laboratories (*Ladhani et al., 2020*). However, every diagnostic test has false-positives, here defining a false-positive as detection of virus by RT-PCR when no virus is present in a sample, so some of our single-gene 'lower', or even 'moderate', evidence positives are inevitably false. However, the false-positive rate (as defined) would generally be expected to be approximately constant over time, since it is either random or driven by external factors, although cross-contamination (which should be minimised by good laboratory practice) may theoretically be related to background prevalence/viral load. Variation in the percentage of all tests accounted for by 'lower' evidence positives, and in particular the proportionate increases in 'lower' evidence positives as 'higher' evidence positives increased during September 2020 supports more genuinely lower-level infections occurring during the summer, and an overall false-positive rate for this test of below ~0.005% that is at least 99.995% specificity.

With recent expansion of antigen assays, there has been considerable debate on what 'positivity' means, and hence what is a 'false-positive' or a 'false-negative'. First, it is clear that the detection of viral RNA is neither the same as infectiousness, although a strong relationship between Ct values and infection in contacts is observed (*Lee et al., 2021*), nor a 'disease' in its own right. However, surveillance has very distinct goals from clinical testing with its focus on isolation and contact tracing, particularly given the large percentage of asymptomatic infections. It is appropriate for surveillance to focus on detection of viral RNA, given its goal to estimate burden of current/ongoing cases that have occurred in the community. However, it is essential to recognise the difference between the RT-PCR test result (viral RNA has been detected) and the appropriate clinical action, which may legitimately differ depending on Ct value, for example if the infection is likely to have occurred sometime previously, as well as other information (e.g. preceding PCR positivity or serology). RT-PCR assays test for viral RNA presence, and hence it is more relevant to consider limits of detection, rather than 'false-positives' per se. Although they were a small minority (6%), one question is whether single-gene positives with high Ct (defined as ≥34 in our study) solely represent long-term shedding of non-transmissible virus (*Moraz et al., 2020*), with, for example, infectious virus recovered from only 8% (95% CI 3–18%) of samples with Ct>35 in a PHE study (*Singanayagam et al., 2020*) and studies reporting no growth of virus for Ct thresholds from >24 to >34 or higher (*Jefferson et al., 2020*). Whilst we have not directly assessed household transmission in this analysis, it was notable that Ct values were significantly lower in positives where anyone else in the same household was ever positive, supporting a role for greater within-household transmission with lower Ct values. Ct values were 0.6 higher in positives that were a participant's first study test (where long-

term shedders would be expected to be overrepresented), but these formed only 14% of the positives.

Our evaluation of serological responses is one of few in the community to our knowledge and highlights that a significant minority (~20%) of RT-PCR-positive cases do not appear to seroconvert, particularly those with higher Ct values and not reporting symptoms. A recent systematic review estimated that 95% of adults with laboratory confirmed SARS-CoV-2 infection developed IgG antibodies (*Arkhipova-Jenkins et al., 2021*), peaking around 25 days. However, only 23% of included studies were in outpatient settings and 14% included only participants with asymptomatic or mild disease. Our community setting, with higher percentages not reporting symptoms and higher Ct values (both associated with not seroconverting), likely explains our lower overall seroconversion estimate compared with these previous studies. We observed a small number of new swab positives in antibody-positive individuals: unfortunately whole-genome sequence data were not available to confirm potential re-infections. Presumed re-infections have been reported elsewhere (*Tomassini et al., 2021*), including in individuals without previous functional and/or durable antibody responses (*Goldman et al., 2020*; *To et al., 2020*), and may remain relevant to virus transmission, whether they occur with or without symptoms. Our data and others (*Lumley et al., 2021*) suggest that these may occur in the presence of anti-spike antibodies, which correlate with neutralising antibody titres. These antibody titres are unlikely to have been false-positives, given the context, persistence, and known diagnostic and analytical specificity of the assay (*National SARS-CoV-2 Serology Assay Evaluation Group, 2020*), or to all reflect laboratory identifier errors, and further analyses are ongoing.

A major study strength is its design, namely being a large-scale community survey recruiting randomly selected private residential households, and testing participants regardless of symptoms. However, its size and scale is also a limitation, since we were not able to collect additional data to comprehensively characterise individual positives. We may have underestimated the initial prevalence of symptoms due to originally asking about current symptoms before July 2020 (subsequently symptoms in the 7 days preceding the visit). As this was only at the earliest visits, mostly weekly, only very transient symptoms between visits would likely have been missed. Similar rates of symptom reporting in the first and last parts of the period analysed suggests that this question was likely generously interpreted in any case. We made no attempt to collect additional information on symptoms after positives were identified to minimise recall bias. This may partly explain why we observed higher rates of positive tests without reported symptoms than recent reviews (*Buitrago-Garcia et al., 2020*; *Byambasuren et al., 2020*); however, many studies in these reviews tested close contacts of index cases identified through symptoms and therefore might plausibly have higher viral loads. We compared distributions of Ct values to overall positivity rates in England, since these are the longest series of official statistics available; overall UK positivity estimates are not produced because the four countries making up the UK have different policies and timings regarding community restrictions including lockdowns.

Ultimately, the importance of asymptomatic and low virus-level infections depends on their transmissibility and their prevalence; regardless of limitations in symptom ascertainment, infection without recognition has the potential for onward transmission and unascertained infections are likely critical for avoiding resurgence after lifting lockdown (*Hao et al., 2020*). Our findings support the use of Ct values and genes detected more broadly in public testing programmes, predominantly testing symptomatic individuals and case contacts, as an 'early warning' system for shifts in potential infectious load and hence transmission, and hence the risks posed by individuals to others. This has recently also been proposed on the basis of theoretical work linking effective reproduction numbers to population-level Ct (*Hay et al., 2020*). In our study, declines in mean and median Ct values preceded or at least coincided with increases in office estimates of positivity rates (*Figure 3B*); given the far larger numbers that would be available in testing programmes, future research should investigate whether the greater power afforded by continuous outcomes could lead to significantly earlier detection of future positivity increases, particularly within small geographical areas. Ct data are widely available within-laboratory management systems; providing comparisons across the wide variety of commercial assays were interpreted carefully, they could be used alongside available risk factor and symptom information to facilitate more informed and effective individual-level and public health responses to the SARS-CoV-2 pandemic.

## Acknowledgements

Office for National Statistics: Iain Bell, Ian Diamond, Alex Lambert, Pete Benton, Emma Rourke, Stacey Hawkes, Sarah Henry, James Scruton, Peter Stokes, Tina Thomas. Office for National Statistics, Analysis John Allen, Russell Black, Heather Bovill, David Braunholtz, Dominic Brown, Sarah Collyer, Megan Crees, Colin Daglish, Byron Davies, Hannah Donnarumma, Julia Douglas-Mann, Antonio Felton, Hannah Finselbach, Eleanor Fordham, Alberta Ipser, Joe Jenkins, Joel Jones, Katherine Kent, Geeta Kerai, Lina Lloyd, Victoria Masding, Ellie Osborn, Alpi Patel, Elizabeth Pereira, Tristan Pett, Melissa Randall, Donna Reeve, Palvi Shah, Ruth Snook, Ruth Studley, Esther Sutherland, Eliza Swinn, Heledd Thomas, Anna Tudor, Joshua Weston. Office for National Statistics, Secure Research Service Shayla Leib, James Tierney, Gabor Farkas, Raf Cobb, Folkert van Galen, Lewis Compton, James Irving, John Clarke, Rachel Mullis, Lorraine Ireland, Diana Airimitoaie, Charlotte Nash, Danielle Cox, Sarah Fisher, Zoe Moore, James McLean, Matt Kerby. University of Oxford, Nuffield Department of Medicine: Ann Sarah Walker, Derrick Crook, Philippa C Matthews, Tim Peto, Emma Pritchard, Nicole Stoesser, Karina-Doris Vihta, Jia Wei, Alison Howarth, George Doherty, James Kavanagh, Kevin K Chau, Stephanie B Hatch, Daniel Ebner, Lucas Martins Ferreira, Thomas Christott, Brian D Marsden, Wanwisa Dejnirattisai, Juthathip Mongkolsapaya, Sarah Cameron, Phoebe Tamblin-Hopper, Magda Wolna, Rachael Brown, Sarah Hoosdally, Richard Cornall, David I Stuart, Gavin Screaton. University of Oxford, Nuffield Department of Population Health: Koen Pouwels. University of Oxford, Big Data Institute: David W Eyre, Katrina Lythgoe, David Bonsall, Tanya Golbchik, Helen Fryer. University of Oxford, Radcliffe Department of Medicine: John Bell. Oxford University Hospitals NHS Foundation Trust: Stuart Cox, Kevin Paddon, Tim James. University of Manchester: Thomas House. Public Health England: John Newton, Julie Robotham, Paul Birrell. IQVIA: Helena Jordan, Tim Sheppard, Graham Athey, Dan Moody, Leigh Curry, Pamela Brereton. National Biocentre Ian Jarvis, Kirsty Howell, Bobby Mallick, Phil Eeles. Glasgow Lighthouse Laboratory Jodie Hay, Harper Vansteenhouse. Department of Health: Jessica Lee. This study is funded by the Department of Health and Social Care. ASW, EP, JVR, TEAP, NS, DE, KBP are supported by the National Institute for Health Research Health Protection Research Unit (NIHR HPRU) in Healthcare Associated Infections and Antimicrobial Resistance at the University of Oxford in partnership with Public Health England (PHE) (NIHR200915). ASW and TEAP are also supported by the NIHR Oxford Biomedical Research Centre. EP and KBP are also supported by the Huo Family Foundation. ASW is also supported by core support from the Medical Research Council UK to the MRC Clinical Trials Unit [MC_UU_12023/22] and is an NIHR Senior Investigator. PCM is funded by Wellcome (intermediate fellowship, grant ref 110110/Z/15/Z) and holds an NIHR BRC Senior Fellowship award. The views expressed are those of the authors and not necessarily those of the National Health Service, NIHR, Department of Health, or PHE. The funders had no role in study design, data collection and interpretation, or the decision to submit the work for publication.

## Additional information

### Competing interests

David W Eyre: declares lecture fees from Gilead, outside the submitted work. The other authors declare that no competing interests exist.

### Funding

| Funder | Grant reference number | Author |
| --- | --- | --- |
| Department of Health & Social Care | - | A Sarah Walker<br>Emma Pritchard<br>Thomas House<br>Iain Bell<br>Ian Diamond<br>Ruth Studley<br>Jodie Hay<br>Karina-Doris Vihta<br>Koen B Pouwels |
| National Institutes of Health | NIHR200915 | A Sarah Walker |

| | | Emma Pritchard |
| | | Julie V Robotham |
| | | Karina-Doris Vihta |
| | | Timothy EA Peto |
| | | Nicole Stoesser |
| | | David W Eyre |
| | | Koen B Pouwels |
| Huo Family Foundation | | Emma Pritchard |
| | | Koen B Pouwels |
| Medical Research Council | MC_UU_12023/22 | A Sarah Walker |
| Wellcome Trust | 110110/Z/15/Z | Philippa C Matthews |

The funders had no role in study design, data collection and interpretation, or the decision to submit the work for publication.

### Author contributions
A Sarah Walker, Conceptualization, Data curation, Formal analysis, Supervision, Methodology, Writing - original draft, Project administration, Writing - review and editing; Emma Pritchard, Jodie Hay, Karina-Doris Vihta, Formal analysis, Writing - review and editing; Thomas House, Koen B Pouwels, Conceptualization, Methodology, Writing - review and editing; Julie V Robotham, Paul J Birrell, Timothy EA Peto, Philippa C Matthews, David W Eyre, Conceptualization, Writing - review and editing; Iain Bell, Conceptualization, Supervision, Funding acquisition, Project administration, Writing - review and editing; John I Bell, John N Newton, Conceptualization, Funding acquisition, Writing - review and editing; Jeremy Farrar, Conceptualization, Funding acquisition, Project administration, Writing - review and editing; Ian Diamond, Funding acquisition, Project administration, Writing - review and editing; Ruth Studley, Conceptualization, Methodology, Project administration, Writing - review and editing; Nicole Stoesser, Conceptualization, Formal analysis, Writing - review and editing; COVID-19 Infection Survey team, Resources; Investigation

### Author ORCIDs
A Sarah Walker https://orcid.org/0000-0002-0412-8509
Paul J Birrell http://orcid.org/0000-0001-8131-4893
Nicole Stoesser https://orcid.org/0000-0002-4508-7969
Philippa C Matthews http://orcid.org/0000-0002-4036-4269
David W Eyre http://orcid.org/0000-0001-5095-6367
Koen B Pouwels https://orcid.org/0000-0001-7097-8950

### Ethics
Human subjects: Written informed consent was obtained from participants aged 16 years and older, and from parents/carers for those aged 2-15 years; those aged 10-15 years provided written assent. The study received ethical approval from the South Central Berkshire B Research Ethics Committee (20/SC/0195).

### Decision letter and Author response
Decision letter https://doi.org/10.7554/eLife.64683.sa1
Author response https://doi.org/10.7554/eLife.64683.sa2

## Additional files
### Supplementary files
• Supplementary file 1. Month of recruitment into the COVID-19 Infection Survey.

• Supplementary file 2. Association between characteristics and Ct values. (A) Univariable effects and main model considering all factors, (B) multivariable model excluding potential mediators of effects of demographics.

• Supplementary file 3. Data points underlying figures.

- Supplementary file 4. Stata code.
- Transparent reporting form

## Data availability

De-identified study data are available for access by accredited researchers in the ONS Secure Research Service (SRS) for accredited research purposes under part 5, chapter 5 of the Digital Economy Act 2017. Individuals can apply to be an accredited researcher using the short form on https://researchaccreditationservice.ons.gov.uk/ons/ONS_registration.ofml. Accreditation requires completion of a short free course on accessing the SRS. To request access to data in the SRS, researchers must submit a research project application for accreditation in the Research Accreditation Service (RAS). Research project applications are considered by the project team and the Research Accreditation Panel (RAP) established by the UK Statistics Authority. Project application example guidance and an exemplar of a research project application are available. A complete record of accredited researchers and their projects is published on the UK Statistics Authority website to ensure transparency of access to research data. For further information about accreditation, contact https://researchaccreditationservice.ons.gov.uk/ons/ONS_homepage.ofml or visit the SRS website. Data points underlying Figures are provided in Supplementary File 4 and Stata code in Supplementary File 3.

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

## Appendix 1

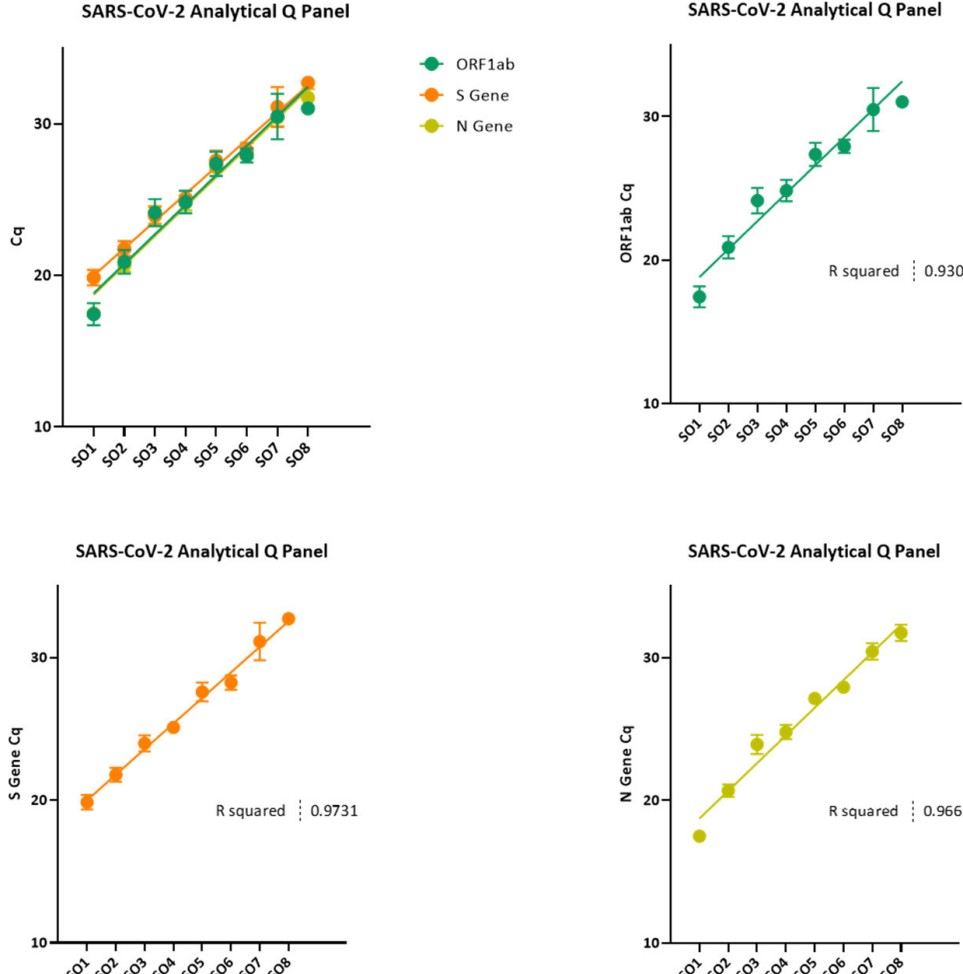

Note: Mean (+/- 95% CI) Ct value plotted against the Qnostics linearity panel (SCV2AQP01-A) with target concentrations from 0 – 1000000 direct copies (dC)/ml (S01=1,000,000 dC/ml, S02=100,000 dC/ml, S03-10,000 dC/ml, S04=5,000 dC/ml, S05=1,000 dC/ml, S06=500 dC/ml, S07=100 dC/ml, S08=50 dC/ml)

**Appendix 1—figure 1.** Relationship between Ct values and viral load.

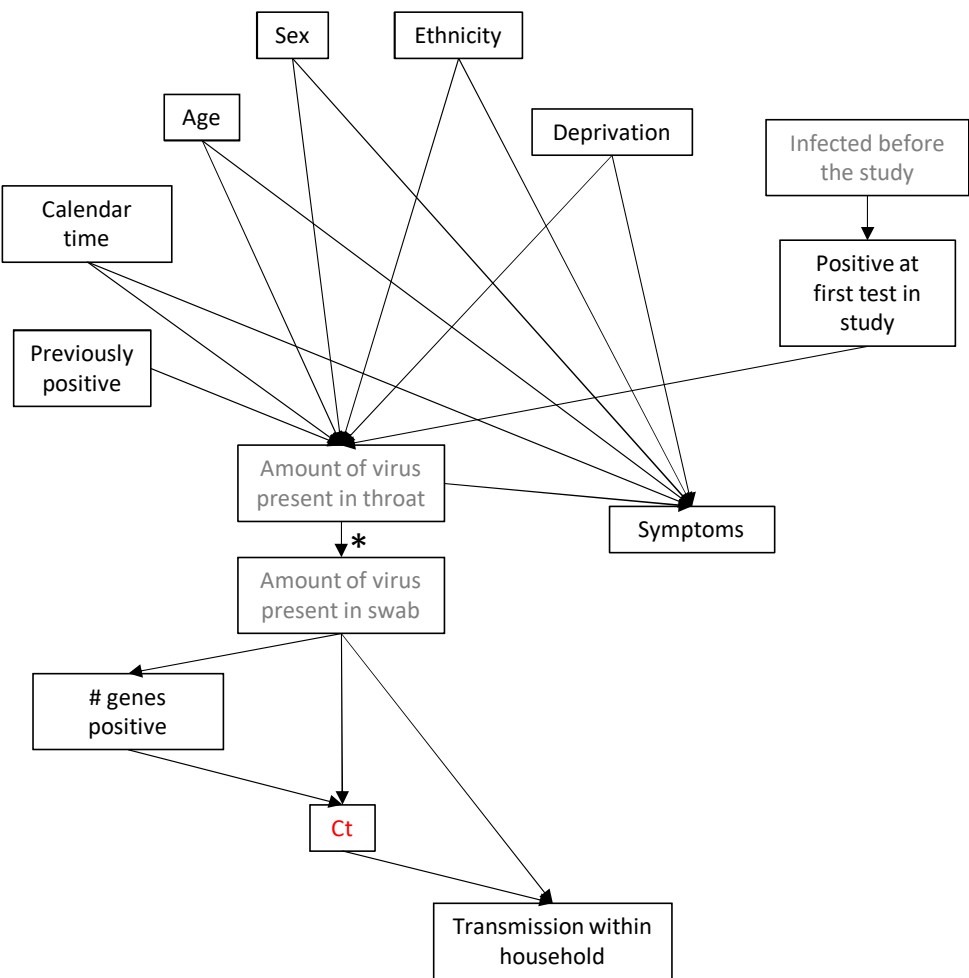

* May also depend on factors which effect self-swabbing efficiency, eg, demographics.
Note: showing hypothesised potential causal relationships between factors that may be associated with the primary outcome, Ct value. Unobserved factors shown in gray, outcome in red.

**Appendix 1—figure 2.** Directed acyclic graph of potential relationships between factors. *May also depend on factors which effect self-swabbing efficiency, e.g., demographics.

