## [Decision Letter]

**Acceptance summary:**

This paper analyses viral levels (detected as CT) in community acquired SARS-CoV-2 swabs. It suggests a relationship with symptoms, as well as suggesting that declines in observed CT might be used as an indicator in increased transmission.

**Decision letter after peer review:**

Thank you for submitting your article "Viral load in community SARS-CoV-2 cases varies widely and temporally" for consideration by *eLife*. Your article has been reviewed by 2 peer reviewers, and the evaluation has been overseen by a Reviewing Editor and Miles Davenport as the Senior Editor. The reviewers have opted to remain anonymous.

Please modify your manuscript to address the comments and recommendations to the authors provided by reviewer 1 and reviewer 2.

*Reviewer #1 (Recommendations for the authors):*

Overall, the paper contains interesting data, with an exciting statement that increasing viral load can be used as an epidemiological early-warning indicator. They also included a nice attempt of a correlation of Ct with clinical evidence to rank the confidence of positive results. The authors also highlight that some of these RT-PCR-positive cases do not appear to seroconvert and reported possible re-infections despite the presence of anti-spike antibodies which given the current situation with SARS-CoV-2 VOC 202012/01 could be of concern.

However, the interpretation of the fluctuating Ct and their assumption that it can be used as an epidemiological early-warning indicator should be better discuss (see major comments) and more importantly demonstrated.

Furthermore, data on Cts are not completely new as some already described high population-level variability of viral loads without strong correlation with disease severity. However, the data presented here are statistically relevant and therefore interesting. Fluctuating Ct were also previously reported to follow epidemic waves and therefore confinement measures. This should be further discussed here. Finally, some formatting work would be required to better present the data especially regarding the serology results which are neglected in the introduction (see comments).

– In the abstract, the author stated: "Cts changed over time, with declining Ct preceding increasing positivity". This is not clearly shown in the paper. It would be interesting to support this claim to present both Ct and positivity on the same graph to demonstrate that indeed, declining Ct can be used as an early marker of an epidemic wave. From the different graphs presented in Figure 1 this is far from evident. As this observation is the main impact statement, the authors should improve their demonstration as this is critically missing. Percentage of positive test data should not only include the ones obtained in the present study but should be compared with "national data" as your study includes a bias in patients’ selection that might not reflect the "true" situation at the time. Only with this comparison, you can claim that your study design could predict epidemic waves and support your impact statement.

– During the ascending phase of a COVID-19 epidemic wave, patients screened harbor mostly high viral loads, reflecting the start of the disease. Indeed, viral load kinetic for SARS-CoV-2 follow simplistically a sharp increase followed by a relatively slow decline over time. Therefore, with confinement measures in place and lower transmission, it is more likely to screen people at multiple stages of the disease and with therefore lower viral loads. The idea to use mean viral loads as early warning of a start of a new wave is an interesting observation and it would be interesting to place that in the perspective of COVID-19 progression within individuals.

– How long before decrease of Ct can predict and epidemic wave?

– Neither the impact statement nor the abstract mention the serological findings presented in the paper. Is this something deliberate?

– The title does not reflect the content of the paper. If demonstrated, it should at least reflect your impact statement.

– Although described in the material and methods, I would include a short description of the UK's national COVID-19 Infection Survey (CIS) in the introduction to help the reader quickly understand the context of the collected specimens described in this study.

– Fluctuating Ct might likely reflect the confinement measures undertaken by the UK government. In this context, it would be relevant to include the main confinement periods imposed by the government on the different graphs and to discuss their relationship with the viral loads.

– The Results section regarding serological data would require some major improvements, as it is difficult to navigate. It would be much easier for the readers to subdivide the figure in panel to better describe some conclusion. For example, "seven participants appeared to have become infected despite antecedent high anti-spike antibody titres (Figure 4)" should be place in a panel to facilitate the reading (the authors included some color-code but it is still difficult to follow).

– All the figure legends would require some improvement to better guide the reader. For example, supplementary figure 2 is cryptic without a proper legend. And again, I recommend the authors to subdivide their figure in panels, with one panel per "conclusion".

– The chronology of the figures is sometimes cryptic with figures appearing in non-chronological order. I would suggest rethinking the flow of the figures.

– In table 2, it is mentioned that multiple genome were sequenced. Is there any correlations between genome vs symptom, genome vs Cts, ? This would add an interesting part to the paper as these data are likely available.*Reviewer #2 (Recommendations for the authors):*

Can the authors add some discussion about that they think the different levels of positive are: high, medium and low. I assume that a true positive is someone in whom we find a fragment of RNA that was made by a cell in their body, even if that was some time ago. A false positive is someone for whom there was no real RNA present but through some kind of amplification error or contamination we say that there was? The results seem to suggest that some of the low confidence positives were actual false positives (changing strength of statistical associations) but I didn't feel that the point was fully made.

Some specific points:

Line 95: Couldn't quite understand the sentence on this line

110: What are the units for viral load?

110: Was there any kind of threshold in the definition of positives? Did the threshold differ for single, double of triple positive samples?

115: What is the positivity definition here? Is a true positive someone who has recently produced SARS-CoV-2 RNA and is still testing positive or is it some higher level? Or, is a true negative someone who has not shed SARS-CoV-2 for at least x weeks?

147: How were symptoms defined? Over what period?

151: What were the effects over calendar time?

188: Viral fitness declined over time: I wasn't quite sure what this would be a hypothesis at this point. Maybe needs setting up a little more clearly.

196: Also worth commenting on the cold chain for sample collection, or other study-related factors. A higher ambient temperature for the samples could have led to faster rates of degradation.

222: There could be a wider point here about household transmission. Chickenpox is known to be more severe in household secondaries than in household index, and CT values seem to be correlated with symptoms (https://pubmed.ncbi.nlm.nih.gov/15702036/)

294: Was there a threshold for the single-gene positives?

301: Is there any clear evidence of how the symptom definitions affected reporting when changing from "current" to "previous seven days".

306: As mentioned above, I think some discussion of false positives is merited in this paper. Do the authors feel that detection of an actual RNA fragment from a recent infection is a true or a false positive? Not sure there is an easy answer here, but with all the discussion of detecting infectious people versus just detecting the virus, a bit more context would help.

Page 30 of merged pdf: there is a multipart figure with antibody values on the y axis that I couldn't find a legend for or a figure number.

Supp Figure 2: shouldn't there be an "average true number of virions in the throat and nose" as an unobserved state? This figure might merit a bit more discussion in the intro and Discussion sections.

---

## [Author Response]

Please modify your manuscript to address the comments and recommendations to the authors provided by reviewer 1 and reviewer 2.Reviewer #1 (Recommendations for the authors):Overall, the paper contains interesting data, with an exciting statement that increasing viral load can be used as an epidemiological early-warning indicator. They also included a nice attempt of a correlation of Ct with clinical evidence to rank the confidence of positive results. The authors also highlight that some of these RT-PCR-positive cases do not appear to seroconvert and reported possible re-infections despite the presence of anti-spike antibodies which given the current situation with SARS-CoV-2 VOC 202012/01 could be of concern.However, the interpretation of the fluctuating Ct and their assumption that it can be used as an epidemiological early-warning indicator should be better discuss (see major comments) and more importantly demonstrated.Furthermore, data on Cts are not completely new as some already described high population-level variability of viral loads without strong correlation with disease severity. However, the data presented here are statistically relevant and therefore interesting. Fluctuating Ct were also previously reported to follow epidemic waves and therefore confinement measures. This should be further discussed here. Finally, some formatting work would be required to better present the data especially regarding the serology results which are neglected in the introduction (see comments).– In the abstract, the author stated: "Cts changed over time, with declining Ct preceding increasing positivity". This is not clearly shown in the paper. It would be interesting to support this claim to present both Ct and positivity on the same graph to demonstrate that indeed, declining Ct can be used as an early marker of an epidemic wave. From the different graphs presented in Figure 1 this is far from evident. As this observation is the main impact statement, the authors should improve their demonstration as this is critically missing. Percentage of positive test data should not only include the ones obtained in the present study but should be compared with "national data" as your study includes a bias in patients’ selection that might not reflect the "true" situation at the time. Only with this comparison, you can claim that your study design could predict epidemic waves and support your impact statement.

We have added the requested panel for England (new Figure 3B) where we have the longest follow-up and the largest number of tests, and have added some further text in the Discussion to support this point – which is also clearer with the additional 6 months data included in the revised manuscript. There are no official statistics on positivity rates for the UK as a whole because the 4 countries comprising the UK (England, Wales, Northern Ireland and Scotland) have different policies on community based restrictions enacted at different times, making a combined estimate challenging to interpret – we have added this point as a limitation in the Discussion to explain our focus on England in Figure 3B. We strongly disagree with the reviewer’s assertion that our study includes a bias in patient selection and that there is “national data” that we should compare to. The COVID-19 Infection Survey is the national data, it is the only continuous surveillance study in the UK.

– During the ascending phase of a COVID-19 epidemic wave, patients screened harbor mostly high viral loads, reflecting the start of the disease. Indeed, viral load kinetic for SARS-CoV-2 follow simplistically a sharp increase followed by a relatively slow decline over time. Therefore, with confinement measures in place and lower transmission, it is more likely to screen people at multiple stages of the disease and with therefore lower viral loads. The idea to use mean viral loads as early warning of a start of a new wave is an interesting observation and it would be interesting to place that in the perspective of COVID-19 progression within individuals.

Whilst we agree with the reviewer that this would be interesting, as noted in the limitation section of the Discussion, the large scale and size of the survey means we do not have this kind of specific level of detail on individual disease progression in those testing positive. Further, as we are testing individuals in the community, the numbers that develop sufficiently severe infections to be admitted to hospital or even die is extremely small.

– How long before decrease of Ct can predict and epidemic wave?

In the new Figure 3B we show that after very low rates in the summer of 2020 the decline in Ct values preceded increases in positivity rates in England. However the steep declines in Ct observed in early December 2020 coincided with very sharp increases in positivity, as the B.1.1.7 variant expanded exponentially. We hypothesise that this very fast expansion explains why the increase followed so quickly. After both periods of increases in positivity rates, rising Ct values were accompanied by stabilisation and then declines in the positivity rate.

– Neither the impact statement nor the abstract mention the serological findings presented in the paper. Is this something deliberate?

Yes, this was deliberate given the limit on word count in the abstract (150 words) and the fact that the original manuscript contained preliminary serological data on a small number of participants. However given that there was at least some information available, we felt that we should provide at least limited results. Given the substantially increased numbers, we have amended this short summary, and now report 78% IgG S-antibody positivity after the first RT-PCR-positive and we have amended the abstract to also include this. Our understanding is that the impact statement should be at most 2 sentences and should focus on consequences and therefore we have still not included it there.

– The title does not reflect the content of the paper. If demonstrated, it should at least reflect your impact statement.

We have amended the title to refer additionally to Ct values which we assume is the reviewer’s point. We do not feel that future implications of our findings are appropriate in the title, although would be happy to reconsider if the Editors preferred this.

– Although described in the material and methods, I would include a short description of the UK's national COVID-19 Infection Survey (CIS) in the introduction to help the reader quickly understand the context of the collected specimens described in this study.

We have added this to the last paragraph of the Introduction as suggested.

– Fluctuating Ct might likely reflect the confinement measures undertaken by the UK government. In this context, it would be relevant to include the main confinement periods imposed by the government on the different graphs and to discuss their relationship with the viral loads.

First, we have had to considerably amend this figures to include over 18,000 additional positives that have been detected since the original submission in October. Second, whilst we are presenting a combined analysis of the features of positives across the UK, as described above lockdown measures are determined independently within each nation (England, Wales, Northern Ireland and Scotland) and therefore there is not one specific lockdown measure across the entire population. In response to the reviewer’s comment, and also in the public review, we have added this information to the new panel B in Figure 3 for England only, presenting the official positivity estimates from the survey together with the Ct values for England and with the “stay-at-home” periods clearly marked in different colours according to whether schools were open or shut.

– The Results section regarding serological data would require some major improvements, as it is difficult to navigate. It would be much easier for the readers to subdivide the figure in panel to better describe some conclusion. For example, "seven participants appeared to have become infected despite antecedent high anti-spike antibody titres (Figure 4)" should be place in a panel to facilitate the reading (the authors included some color-code but it is still difficult to follow).

Given the far larger number of positive tests in the study we have completely rewritten this paragraph to provide a much shorter overall summary of the data to date, with a new main Figure 5 summarising the percentage of positive antibody tests over time before and after the first RT-PCR positive swab. Additional more detailed analyses are ongoing and will be presented separately.

– All the figure legends would require some improvement to better guide the reader. For example, supplementary figure 2 is cryptic without a proper legend. And again, I recommend the authors to subdivide their figure in panels, with one panel per "conclusion".

We have added a footnote to Supplementary Figure 2 (now Appendix 1 Figure 2) as suggested, and revisited all the figures, also in the light of the additional data included.

– The chronology of the figures is sometimes cryptic with figures appearing in non-chronological order. I would suggest rethinking the flow of the figures.

We have reordered the figures as suggested; these have also altered somewhat given the substantially larger numbers of positives now included, also spanning the rise and expansion of B.1.1.7.

– In table 2, it is mentioned that multiple genome were sequenced. Is there any correlations between genome vs symptom, genome vs Cts, ? This would add an interesting part to the paper as these data are likely available.

In this analysis, we used whole genome sequencing as an independent confirmation of presence of virus, and have clarified this in the table. Investigating associations between genetic features and outcomes such as symptoms and Ct requires GWAS approaches; as sequences are available for only the minority of positives, and the paper is already relatively lengthy, we consider that this would be best done separately.

Reviewer #2 (Recommendations for the authors):Can the authors add some discussion about that they think the different levels of positive are: high, medium and low. I assume that a true positive is someone in whom we find a fragment of RNA that was made by a cell in their body, even if that was some time ago. A false positive is someone for whom there was no real RNA present but through some kind of amplification error or contamination we say that there was? The results seem to suggest that some of the low confidence positives were actual false positives (changing strength of statistical associations) but I didn't feel that the point was fully made.Some specific points:Line 95: Couldn't quite understand the sentence on this line.

We have reworded to try to improve clarity and removed the abbreviation.

110: What are the units for viral load?

DC indicated direct copies from the QA panel, we have replaced with copies/ml in the main text and defined in Appendix 1 Figure 1 footnote.

110: Was there any kind of threshold in the definition of positives? Did the threshold differ for single, double of triple positive samples?

As described in the Methods, positivity for each gene is determined by the UgenTec Fast Finder 3.300.5 algorithm, which is FDA accredited. We have clarified both here and in the Methods that there is no specific Ct threshold for determining positivity.

115: What is the positivity definition here? Is a true positive someone who has recently produced SARS-CoV-2 RNA and is still testing positive or is it some higher level? Or, is a true negative someone who has not shed SARS-CoV-2 for at least x weeks?

We have clarified here that we are talking about genuine presence of virus, which we would argue is the most relevant measure for a surveillance study. We have also added further Discussion on this important point, in response to the reviewer’s comment below.

147: How were symptoms defined? Over what period?

This information is provided in the Methods; as symptoms are referred to at multiple places in the Results before this, we have added a reference to the Methods the first time this occurs.

151: What were the effects over calendar time?

These are given in the second part of this sentence “with markedly fewer positives with Ct <30 (Figure 1B), very low percentages with symptoms at/around positive tests, and more “lower” evidence positives in July/August” – the point being made is that Ct, symptoms and level of evidence all follow similar patterns over calendar time. (Note this figure is now Figure 3, having reordered as suggested by the first reviewer above.)

188: Viral fitness declined over time: I wasn't quite sure what this would be a hypothesis at this point. Maybe needs setting up a little more clearly.

We have clarified this as suggested – at least in the UK, numerous individuals postulated that declining positivity during July/August 2020 was a consequence of declining viral fitness as the virus adapted to human hosts.

196: Also worth commenting on the cold chain for sample collection, or other study-related factors. A higher ambient temperature for the samples could have led to faster rates of degradation.

We have noted this point here in the Discussion, and also added details on the transport of swabs from household to laboratory to the Methods.

222: There could be a wider point here about household transmission. Chickenpox is known to be more severe in household secondaries than in household index, and CT values seem to be correlated with symptoms (https://pubmed.ncbi.nlm.nih.gov/15702036/)

A separate analysis specifically investigating household transmission is about to be submitted. Given the length of the existing manuscript, we would prefer not to address this further here.

294: Was there a threshold for the single-gene positives?

No, there is no specific threshold for determining positivity (clarified in the Methods). The UgenTec Fast Finder 3.300.5 algorithm incorporates multiple aspects of the individual Ct curves to determine positivity.

301: Is there any clear evidence of how the symptom definitions affected reporting when changing from "current" to "previous seven days".

This information is implicitly presented in Figure 3C (was 1C in the original submission) which shows percentage of positives reporting symptoms over calendar time. Symptom reporting dropped markedly towards the end of June 2020 concurrent with declines in Ct, the questionnaires were changed mid July 2020, and then symptoms rose markedly from mid-August concurrent with increases in Ct. The challenge therefore is that the questionnaire change is confounded with changing Ct over calendar time. As the vast majority of data in the revision is after July 2020 we have not elaborated further on this.

306: As mentioned above, I think some discussion of false positives is merited in this paper. Do the authors feel that detection of an actual RNA fragment from a recent infection is a true or a false positive? Not sure there is an easy answer here, but with all the discussion of detecting infectious people versus just detecting the virus, a bit more context would help.

As suggested, we have expanded our discussion on this in the paragraph originally starting “Since RT-PCR and antigen assays test for viral presence, it is more relevant to consider limits of detection, rather than “false-positives” per se”. We would argue that it is essential to distinguish the difference between a test result and the action arising from the test result, and also the context of the study – for surveillance our goal is not detecting infectious people in the same way that symptomatic testing (with subsequent isolation, contact tracing etc) needs to. Therefore there are different “gold standards”.

Page 30 of merged pdf: there is a multipart figure with antibody values on the y axis that I couldn't find a legend for or a figure number.

All figures are now provided as separate tif files including Figure titles and legends.

Supp Figure 2: shouldn't there be an "average true number of virions in the throat and nose" as an unobserved state? This figure might merit a bit more discussion in the intro and Discussion sections.

We have added this additional unobserved state to Supplementary Figure 2 (now Appendix 1 Figure 2) as suggested, and also added a paragraph to the Discussion referring to this figure and the updated related findings.